# Impact of Language Guidance: A Reproducibility Study

## Abstract

Modern deep-learning architectures need large amounts of data to produce state-of-the-art results. Annotating such huge datasets is time-consuming, expensive, and prone to human error. Recent advances in self-supervised learning allow us to train huge models without explicit annotation. Contrastive learning is a popular paradigm in self-supervised learning. Recent works like SimCLR and CLIP rely on image augmentations or directly minimizing cross-modal loss between image and text. Banani et al. (2023) propose to use language guidance to sample view pairs. They claim that language enables better conceptual similarity, eliminating the effects of visual variability. We reproduce their experiments to verify their claims and find that their dataset, RedCaps, contains low-quality captions. We use an off-the-shelf image captioning model, BLIP-2, to replace the captions and improve performance, and we also devise a new metric to evaluate the semantic capabilities of self-supervised models based on interpretability methods.

## 1 Introduction

Deep learning thrives on large datasets and compute-intensive training. While unlabeled data is abundant, supervised learning algorithms require annotated data. Annotation of huge datasets is prohibitively expensive, labour-intensive, and error-prone. Self-supervised learning (SSL) enables the model to learn rich and transferable representations from unlabeled data. This has unlocked new possibilities in both computer vision (Chen et al., 2020a; Caron et al., 2021) and natural language processing (Devlin et al., 2018).

Contrastive learning is a self-supervised learning technique in which a model is trained to bring similar images near by in embedding space while pushing dissimilar images far away. SimCLR (Chen et al., 2020a) uses image augmentations such as random crop, Gaussian blur, and random flipping to generate a positive pair while treating other images as negative samples. Other methods (Caron et al., 2018; Wu et al., 2018) use clustering algorithms or nearest neighbour operations to find positive samples. These methods only use visual similarity to find similar images. Two objects might be visually similar, while objects of the same class might be visually dissimilar. In contrast to this, conceptually similar images are more often described similarly. This suggests that leveraging language modality can improve contrastive learning.

Radford et al. (2021) proposes learning a joint embedding space for images and their captions. This yields highly generalizable and accurate representations. However, Banani et al. (2023) suggests that combining embedding spaces of different modalities might lead to sub-optimal results. They propose a new sampling procedure for contrastive learning where image pairs are sampled using caption similarity based on embeddings generated using a pre-trained language encoder.

Banani et al. (2023) retrain existing self-supervised visual learning architectures (Chen et al., 2020a; Caron et al., 2018; Wu et al., 2018) with the proposed sampling strategy. Their experiments show that the newly proposed method outperforms all baselines on varying downstream tasks across multiple datasets. This substantiates the claim that language is a good proxy for conceptual similarity.

We aim to rigorously evaluate these claims by closely replicating the experimental setup and results reported in the original paper. We identify the poor caption quality of the dataset(Desai et al., 2021) used by the original authors and generate better captions from an off-the-shelf caption generator (Li et al., 2023) and

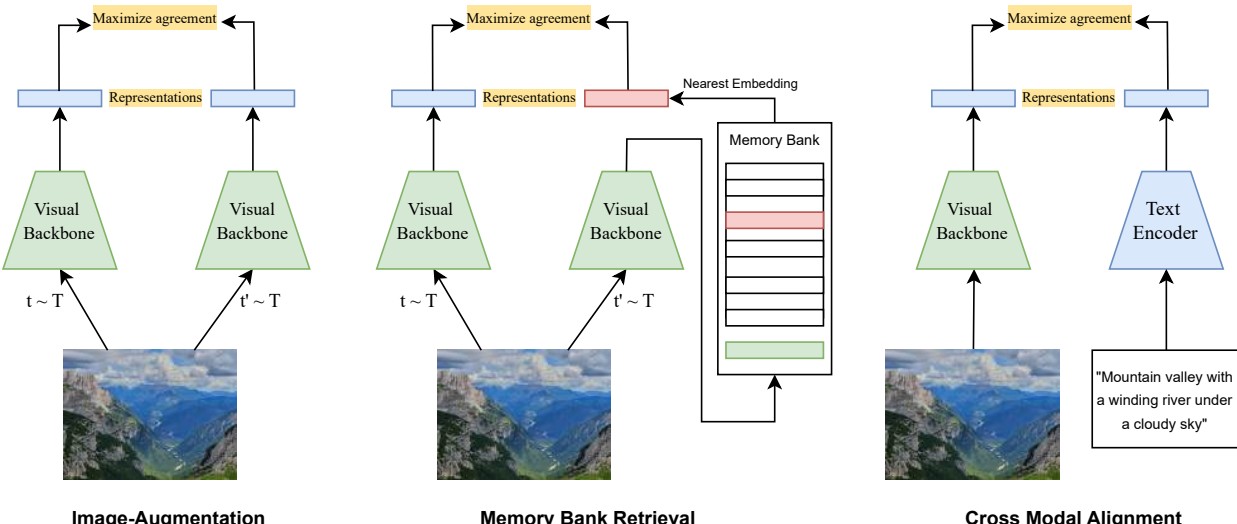

Figure 1: **Refining Representation Learning.** While early contrastive learning methods relied on simple image transformations, newer structured retrieval techniques have emerged to refine the learned embeddings beyond instance-level. These employ clustering, memory banks, or language-driven sampling to introduce structure into training signals and subsequently improve visual representation learning.

analyze the performance improvement. We also demonstrate that the model learns semantic information using interpretability methods (Selvaraju et al., 2017).

## 2 Scope for Reproducibility

The main contribution of the original paper is a new language-based sampling strategy, and their claim is that this strategy improves the underlying self-supervision framework (Banani et al., 2023).

In an effort to reproduce the paper and gain a deeper understanding, we discovered several key limitations:

- **Inefficient Captions:** The method's efficacy is heavily dependent on image captions. Manual inspection of the dataset, RedCaps, reveals that the captions scraped from Reddit are often noisy, vague and inaccurate potentially hampering the model training process.

- **RedCaps Dependency:** The method relies on the availability of captions for the images. It achieves optimal performance when pairs are sampled from specific subreddit subsets. This introduces weak supervision (Zheng et al., 2021). The dataset-specific constraints reduce the generalizability of the method.

- **High Computational Requirements:** The reported results utilized ResNet-50(He et al., 2016) with a batch size of 512 trained from scratch, which requires substantial computational resources that may not be readily available in many settings.

**Our Contributions** To address these limitations and extend the work, we make the following contributions:

- **Reproducibility:** We provide an exhaustive replication of most experiments in the original paper, adapted for reduced computational environments.

- **Visual Backbone Optimization:** We investigate the generalization capabilities of language-guided SSL to smaller, more efficient architectures such as ResNet34 (Wightman et al., 2021) and MobileNetV3 (Howard et al., 2019), making the approach accessible within resource constraints.

- **Caption Quality Enhancement:** We develop a curated set of refined captions for the existing dataset (Desai et al., 2021), improving the impact of language guidance and reducing dataset dependency and enabling generalization to diverse datasets.

- **New Metric:** We generate saliency maps(Selvaraju et al., 2017), which are used to create a new metric for evaluating SSL-trained ConvNets.

## 3 Background

In this section, we provide an overview of the foundational works and existing research that have contributed to advancements in this field. We discuss relevant literature, methodologies, and key developments that form the basis of our study, highlighting their significance in the context of our work.

**Visual Representation Learning** involves learning to encode visual information in an embedding space that preserves its semantics well. Unlike typical machine learning tasks like classification or segmentation, we cannot manually annotate ground truth labels for this task. So, we cannot directly optimize loss in embedding space. Two main approaches have been explored for this task: generative and discriminative. Generative approaches (Doersch et al., 2015; Gidaris et al., 2018; Oord et al., 2018; Vincent et al., 2008; Zhang et al., 2016) involve learning a model that can capture image distribution well. Such models are hypothesized to learn semantically relevant features. Discriminative approaches involve learning a model that can differentiate between images. Understanding semantically relevant features is essential for excelling in tasks like metric learning (Chopra et al., 2005), dimensionality reduction (Hadsell et al., 2006) and classification (Sharif Razavian et al., 2014). Self-supervised learning has recently gained popularity as a visual representation learning method. They relieve the need for human annotation and allow learning from large, unlabelled data sources. Various contrastive (Wu et al., 2018; Chen et al., 2020a;b;c; He et al., 2020), and non-contrastive (Chen & He, 2021; Grill et al., 2020) approaches have been proposed recently. (Banani et al., 2023) propose a sampling strategy for contrastive visual representation learning.

**Image-Image Contrastive Learning.** Contrastive learning involves learning an embedding space in which similar images are close and dissimilar images are far away. Sampling positive and negative pairs effectively is an essential task for the effectiveness of contrastive learning. Chen et al. (2020a) propose a framework called SimCLR for contrastive learning. It involved using data augmentation to generate positive samples for each instance while treating all other images in the batch as negative samples. SimSiam, proposed by (Chen & He, 2021), uses a similar approach but does not use negative pairs during training. It also implements a stop gradient operation in one branch of the Siamese network. It claims that the stop gradient operation is essential to prevent model collapse. Caron et al. (2020) propose SwAV, a new method which reduces computational complexity as it does not need to calculate explicit pairwise comparisons. It relies on an online clustering approach. They compute the feature of an image and then compute its code by matching it with a set of $k$ prototype vectors. Then, they predict the code from an augmented view of the image. These approaches mitigate the demand for a memory bank and reduce computational complexity, ignoring similarities between different instances. NNCLR, proposed by Dwibedi et al. (2021), utilizes similarity between different instances along with transformations to account for more semantic variation. It uses a memory bank similar to He et al. (2020) and samples the nearest neighbour of the image in latent space. It then minimizes the loss between the nearest neighbour and random augmentation of the original image.

**Using language for contrastive learning** Past work has aimed to learn joint vision-language representations for tasks like Visual Question Answering (Antol et al., 2015; Goyal et al., 2017; Hudson & Manning, 2019; Zhu et al., 2016), Visual Reasoning (Kazemzadeh et al., 2014; Suhr et al., 2019; Zellers et al., 2019) and Retrieval (Park et al., 2022; Young et al., 2014). Radford et al. (2021) introduced CLIP, which learned a joint vision-language embedding space by directly minimizing cross-modal loss. This approach was widely adopted due to its generalization and few shot capabilities. Other work improved upon this by adding additional losses and other improvements (Jia et al., 2021; Xu et al., 2022; Yao et al., 2022; Cui et al., 2022; Lee et al., 2022; Li et al., 2022; Mu et al., 2022).

Banani et al. (2023) aims to achieve good results in image-image contrastive learning with the help of language to sample positive pairs.

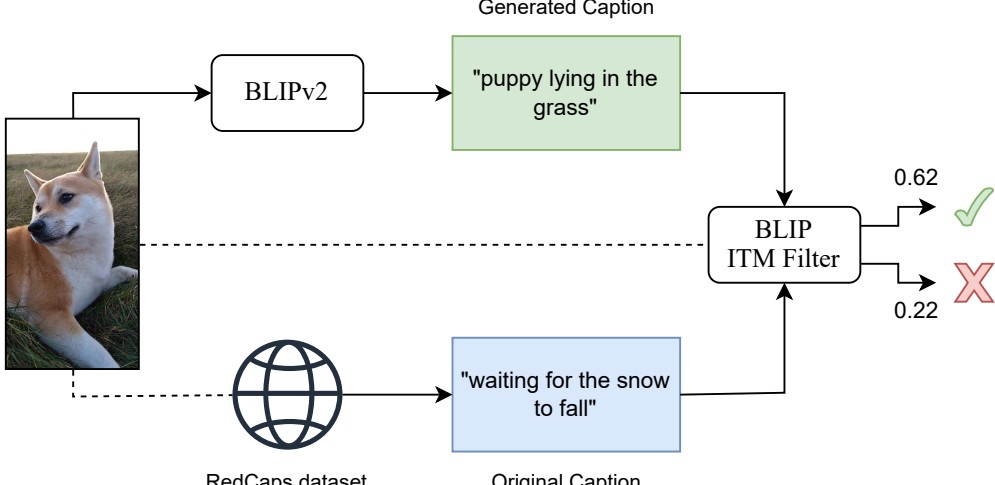

Figure 2: **Improving Captions via Contrastive Filtering.** Our caption improvement method leverages BLIPv2 to generate candidate captions that better describe an image. Since dataset-provided captions can be less relevant or inaccurate, we introduce an Image-Text Matching (ITM) Filter to evaluate, assign a relevance score and select the most appropriate caption between of the two. This ensures better semantic and visual alignment of a caption with its corresponding image.

## 4    Methodology

Traditional SSL methods rely on image augmentations like cropping, colour jitter and noise addition to generate positive pairs. The underlying hypothesis is that visual similarity mirrors semantic similarity. While these methods learn features invariant to specific augmentations, they fail to capture semantic similarity well.

Our reproducibility study examines a language-guided sampling strategy that uses textual captions to identify semantically similar images. The hypothesis is that similar captions capture shared conceptual content beyond what visual augmentations can provide.

Most SSL frameworks use visual backbones like ConvNets and ViTs (Caron et al., 2021) to learn visual representations. Banani et al. (2023) use ResNet50 as the visual backbone for their experiments. To test the generalizability of the framework we performed all experiments for ResNet34. Although both have similar architectures and have roughly the same number of parameters, 25.5M and 21.8M, respectively, they differ significantly in terms of their feature embedding size, 2048 and 512, respectively.

### 4.1    Pair Sampling

Banani et al. (2023) use RedCaps (Desai et al., 2021), a dataset scraped from Reddit. It consists of images with user-written captions. They make image pairs by comparing the similarity of their corresponding caption embeddings. The embeddings are generated using SBERT (Reimers & Gurevych, 2019) and the two captions with the greatest cosine similarity are paired.

Metrics like BLEU (Papineni et al., 2002) and CIDER (Vedantam et al., 2015) are n-gram-based methods traditionally used to find caption similarity. However, n-gram-based approaches are sensitive to variations in phrasing and sentence structure. Though SPICE (Anderson et al., 2016) uses parse trees and handles structural variations better, it is still limited in dealing with different word choices for the same concept. The choice of right caption similarity metric is foundational to the success of their method. SBERT effectively identifies semantically similar caption pairs while being robust to surface-level text variations. The use of cosine similarity simplifies the implementation while maintaining reliable semantic matching capabilities. It is observed that the method is agnostic to the sentence encoder chosen.

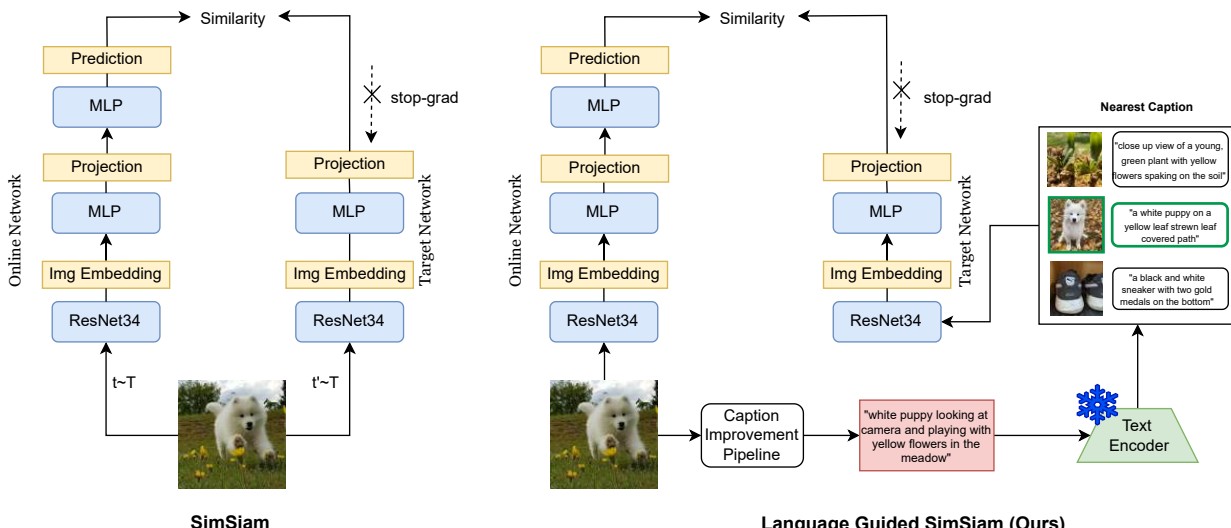

Figure 3: A schematic comparison of SimSiam and Language Guided SimSiam trained using our pipeline.

The FAISS algorithm (Johnson et al., 2019) is used to find the nearest neighbour for a caption in language embedding space. The image corresponding to the most similar caption is chosen as a positive sample for the original image. The similarity search took approximately 21 minutes to complete on our system.

## 4.2 Improving the Dataset

We identified that the quality of Reddit-sourced captions could be a significant limiting factor. The RedCaps dataset, while extensive, contains captions that are often vague, noisy, and inconsistent in their descriptive quality. To test this hypothesis and potentially improve the method, we introduced BLIPv2 as an alternative caption generation approach. BLIPv2 generates concise, descriptive captions that maintain consistent quality across the dataset. We adopt a filtering strategy where we generate new captions using BLIPv2 and evaluate their quality using Image-Text Matching (ITM) scores. The ITM score was developed by Li et al. (2023). It is obtained using a multimodal transformer-based binary classifier, which is trained to classify whether an image-text pair matches or not. Higher ITM scores indicate captions that are more semantically relevant to the image. Captions with higher ITM scores are retained to ensure better language guidance for contrastive learning, as shown in Figure 2. The strategy can also be adopted in cases where captions are unavailable. We generate a single caption per image and compare it to the original. However, multiple captions can also be sampled using BLIP, and the caption with the highest ITM score can be retained.

Our modification serves two key purposes. Firstly, it allows us to evaluate whether higher-quality captions improve the performance of language-guided SSL. Secondly, it enables language guidance frameworks to be extended to any image dataset, regardless of whether it contains associated text descriptions.

## 4.3 Visualizing learned representations

We use self-supervised learning methods to learn visual representations. These representations are used for downstream tasks and evaluated on it. However, it is essential to inspect if the model is focussing on the right regions of the image. We train a linear probe on the ResNet-34 backbone using the train set of ImageNetS-50. We apply GradCAM (Selvaraju et al., 2017) on the second convolutional block of layer 4 of the backbone with true class of the image.

## 5 Experimental Setup

We primarily base our experiments on the code[1] provided by the authors. Our experimental evaluation focuses on thoroughly validating the impact of improved captions on self-supervised learning frameworks. We explore multiple frameworks while maintaining consistent training conditions across all experiments to ensure fair comparisons.

To comprehensively evaluate the effect of improved captions, we conduct experiments across a diverse set of self-supervised learning frameworks. Our study includes SimCLR, LGSimCLR, SimSiam, SwAV, and NNCLR. This selection enables us to verify whether the performance improvements from better captions generalize across different architectural approaches, as demonstrated in Table 2.

### 5.1 Training Details

We perform all experiments using a ResNet-34 backbone trained across on two NVIDIA V100s. To ensure meaningful comparisons across different experimental conditions, we maintain consistent training parameters throughout our studies. We employ the AdamW optimizer (Loshchilov & Hutter, 2016) with the same hyperparameters used by Banani et al. (2023): learning rate of 0.001 and weight decay of 0.01. The learning schedule follows a cosine decay pattern with 5000 warm-up steps. Each model is trained for 25 epochs with a batch size of 512 images resulting in a training time of approximately 1.5 hours per epoch.

We choose RedCaps-2020 as our training dataset. It is a subset of the RedCaps dataset comprising 2.8 million image-text pairs uploaded on Reddit in 2020. We explore two distinct caption sources in our experiments. First, we establish baseline performance using the original RedCaps captions. Then, we use our caption enhancement pipeline to measure the direct impact of caption quality on model performance.

### 5.2 Evaluation Protocol

We run different downstream tasks on the frozen features for each model across multiple datasets to evaluate their performance. Similar to the original authors, we evaluate the model on linear-probe classification (Kornblith et al., 2019) and few-shot classification (Wang et al., 2019). We were able to reproduce results for all datasets mentioned in the original paper except Sun397, Cars, Caltech-101 and Oxford Flowers. The Sun397 dataset has several corrupted images; Cars dataset has been removed from the host site; and the authors' code implementation to download Caltech-101 and Oxford Flowers is not working. Additionally, we report results using a new approach to evaluate self-supervised models using saliency maps.

| Metric | SimCLR | LGSimCLR | LGSimCLR (*Ours*) |
|---|---|---|---|
| AUC-ROC | 0.5411 | 0.5195 | **0.5501** |
| AUC-PR | **0.3419** | 0.3244 | 0.3416 |

Table 1: Saliency Map Evaluation. We evaluate across three models i.e. SimCLR, LGSimCLR on original captions and LGSimCLR on new captions (Ours) across two metrics. All the models were retrained with ResNet-34 backbone.

**Saliency Map Evaluation** We generate saliency maps using the method described in subsection 4.3. We evaluate the saliency maps using the Area Under the Precision-Recall Curve (AUC-PR) and the Area Under the ROC Curve (AUC-ROC) (Cong et al., 2018). These are calculated by treating the ground truth segmentation map as targets for pixel-level classification. We use the validation set of ImageNet-S50 for evaluation. Results are reported in Table 1. We can see that the metrics are not significantly different. So, language guidance does not seem to have a significant impact on the quality of saliency maps. We visualize saliency maps for all validation images and notice that each model performs better than others on some categories while performing worse on others. We display this for 3 out of 50 categories in Figure 4.

---

[1]https://github.com/mbanani/lgssl

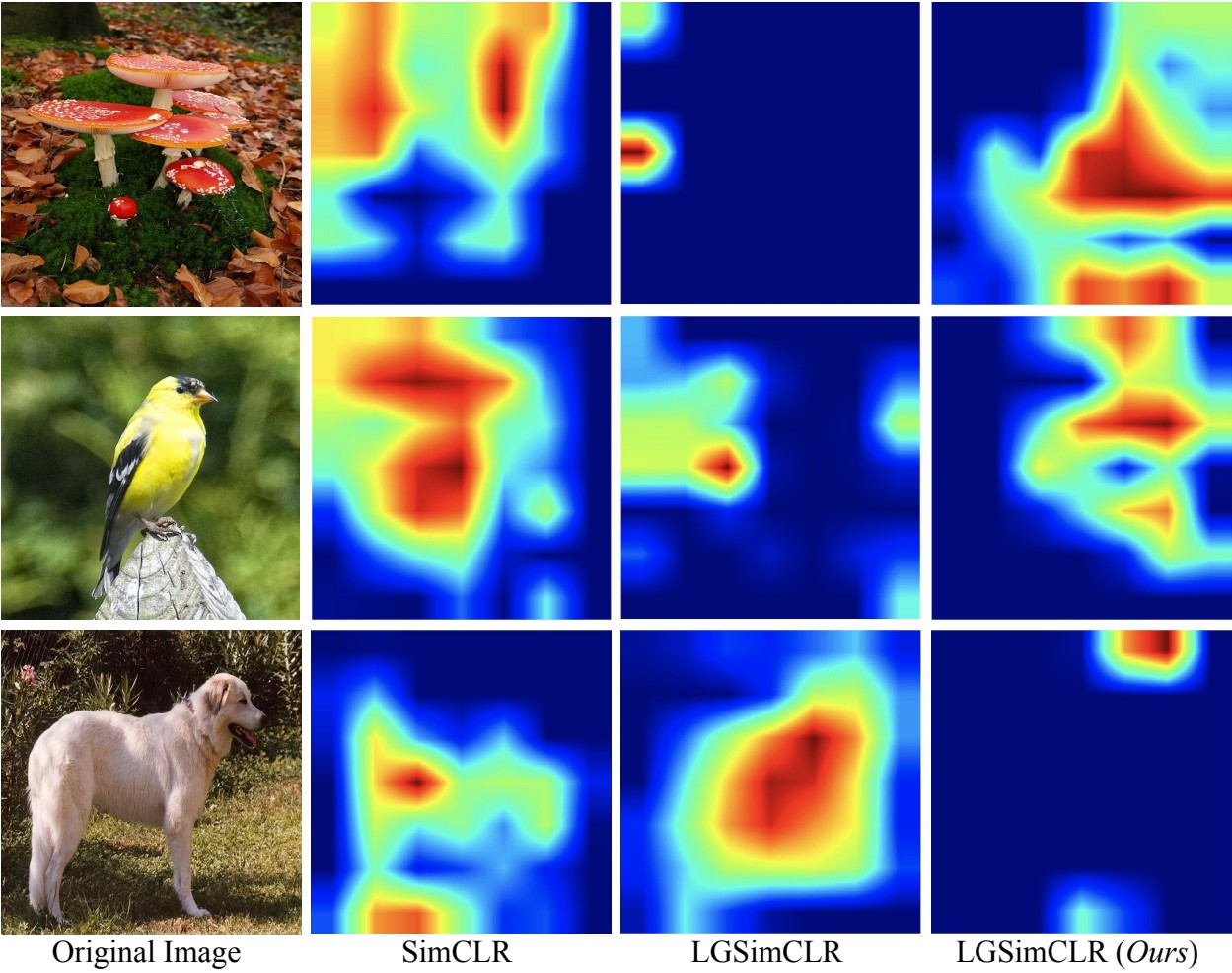

| Original Image | SimCLR | LGSimCLR | LGSimCLR (*Ours*) |

Figure 4: **Visualisations.** Different models perform better on different classes. **Top**: LGSimCLR (*Ours*) performs the best. **Middle**: SimCLR performs the best. **Bottom**: LGSimCLR performs the best.

## 6    Results and Discussion

We report the classification results in Table 2, Table 3 and Table 4. Our experiments suggest that the impact of language guidance is not as profound as indicated in the original paper. The performance disparity between ResNet34 and ResNet50 may be largely attributed to differences in the size of their feature embedding spaces since their architecture and overall parameter counts are comparable, 21.79M and 25.5M respectively. ResNet50 produces a 2048-dimensional feature representation in contrast to the 512 dimensional feature representation of ResNet34. The large embedding space of ResNet50 likely allows the network to capture a rich set of features. Conversely, the reduced capacity of a 512-dimensional embedding space may limit the model's ability to fully exploit the semantic cues provided by the language guidance, resulting in a less pronounced improvement in performance. Neglecting this factor might have led to an overestimation of the generalizability of this method. We also ran experiments using MobileNetV3 as the backbone. The classification results were near-random across all models. We hypothesise that this was either due to the significantly small model size, 5.5M parameters, or smaller embedding space of 1024 dimensions.

Our caption improvement pipeline performs better than Banani et al. (2023) in most cases. It boosts LGSimSiam's performance on the 10 fine-grained datasets and few-shot evaluation on ImageNet but fails to show improvement on linear probe evaluation on ImageNet. Similarly, our pipeline improves SimCLR's performance on ImageNet and few-shot evaluation on the fine-grained datasets but slightly worsens the

performance on linear probe evaluation for the fine-grained dataset. This inconsistency could be attributed to the much-reduced output embedding size of the ResNet34 causing stochasticity in results.

Visual baseline models showed a gradual improvement till the end of the training. In contrast, the language-guided counterparts began to overfit soon, hence, performance significantly worsened as shown in Figure 6 and Figure 5. Based on these observations, we report results at the epoch with the highest validation performance for language guided SimCLR and SimSiam at the $9^{th}$ and $6^{th}$ epoch, respectively.

| Models | Food101 | CIFAR10 | CIFAR100 | CUB | Aircraft | DTD | Pets | STL10 | EuroSAT | RESISC45 | Avg |
|---|---|---|---|---|---|---|---|---|---|---|---|
| NNCLR | 52.7 | 69.9 | 45.4 | 17.3 | 25.7 | 59.2 | 56.8 | 83.6 | 91.8 | 71.8 | **57.42** |
| SWAV | 50.7 | 69.0 | 43.0 | 15.3 | 23.2 | 57.6 | 54.1 | 83.2 | 90.4 | 68.6 | 55.51 |
| *SimCLR* | | | | | | | | | | | |
| Visual | 61.2 | 77.5 | 53.0 | 27.3 | 36.0 | 61.9 | 66.4 | 85.1 | 94.2 | 78.7 | **64.13** |
| Banani et al. | 66.7 | 72.9 | 51.1 | 38.5 | 32.4 | 59.3 | 65.5 | 84.2 | 89.8 | 77.5 | 63.79 |
| Ours | 64.4 | 75.0 | 51.2 | 33.4 | 30.7 | 59.5 | 67.8 | 87.3 | 91.1 | 77.2 | 63.76 |
| *SimSiam* | | | | | | | | | | | |
| Visual | 56.0 | 72.0 | 46.1 | 21.1 | 27.9 | 58.6 | 55.9 | 85.2 | 92.3 | 72.4 | 58.75 |
| Banani et al. | 54.9 | 71.9 | 49.8 | 23.5 | 30.7 | 52.6 | 46.9 | 79.0 | 90.1 | 75.1 | 57.45 |
| Ours | 55.8 | 73.8 | 50.8 | 23.6 | 28.6 | 56.1 | 53.4 | 85.7 | 90.2 | 74.8 | **59.28** |

Table 2: **Linear Probe Classification.** We report the performance of a linear probe using frozen features on 10 downstream tasks. The first split refers to models that language guidance is not applicable to. The second and third splits refer to different versions of SimCLR and SimSiam respectively.

| Models | Food101 | CIFAR10 | CIFAR100 | CUB | Aircraft | DTD | Pets | STL10 | EuroSAT | RESISC45 | Avg |
|---|---|---|---|---|---|---|---|---|---|---|---|
| NNCLR | 64.0 | 50.9 | 56.4 | 45.1 | 33.8 | 70.7 | 71.0 | 74.8 | 75.2 | 69.3 | **61.12** |
| SWAV | 64.5 | 51.6 | 55.8 | 44.0 | 33.9 | 71.2 | 69.6 | 74.3 | 72.3 | 68.5 | 60.57 |
| *SimCLR* | | | | | | | | | | | |
| Visual | 67.8 | 52.9 | 59.5 | 54.7 | 41.5 | 74.6 | 73.6 | 73.9 | 82.0 | 77.5 | 65.80 |
| Banani et al. | 82.5 | 51.6 | 63.3 | 73.4 | 42.7 | 73.5 | 76.2 | 75.2 | 76.3 | 82.5 | 69.72 |
| Ours | 80.2 | 56.8 | 66.1 | 68.8 | 39.5 | 72.6 | 78.1 | 80.8 | 78.5 | 81.3 | **70.27** |
| *SimSiam* | | | | | | | | | | | |
| Visual | 61.2 | 51.4 | 56.6 | 43.6 | 33.5 | 72.1 | 62.9 | 73.1 | 75.2 | 68.6 | 59.82 |
| Banani et al. | 70.5 | 52.9 | 62.4 | 51.4 | 38.6 | 69.6 | 55.7 | 70.1 | 76.6 | 79.0 | 62.68 |
| Ours | 72.4 | 54.5 | 64.8 | 53.2 | 36.8 | 70.3 | 61.2 | 78.3 | 77.9 | 78.7 | **64.81** |

Table 3: **Few Shot Classification.** We report the performance of 5-way, 5-shot classification using frozen features on 10 downstream tasks. LGSimCLR-*Ours* outperforms previous approaches on most datasets.

| | Without language guidance | | | | Original Captions | | Our Captions | |
|---|---|---|---|---|---|---|---|---|
| | SimCLR | SimSiam | SwAV | NNCLR | LGSimSiam | LGSimCLR | LGSimSiam | LGSimCLR |
| Linear | 68.5 | 65.2 | 72.0 | 66.1 | 65.8 | 67.0 | 65.4 | **72.7** |
| Few-shot | 82.2 | 80.3 | 81.2 | 80.9 | 83.7 | 87.8 | 84.5 | **88.1** |

Table 4: Performance comparison of different models using linear and few-shot evaluation on ImageNet.

# 7   Communication with Authors

Banani et al. (2023) conducted experiments on data subsampled from different subreddits. The code provided by them corresponding to these experiments was incomplete, lacking invocation of functions or definitions of functions themselves. We raised issues on their GitHub repository as it was unclear how to complete the code implementation. Attempts to resolve these roadblocks by raising issues in the author's GitHub repository were in vain, as the issues remained unaddressed.

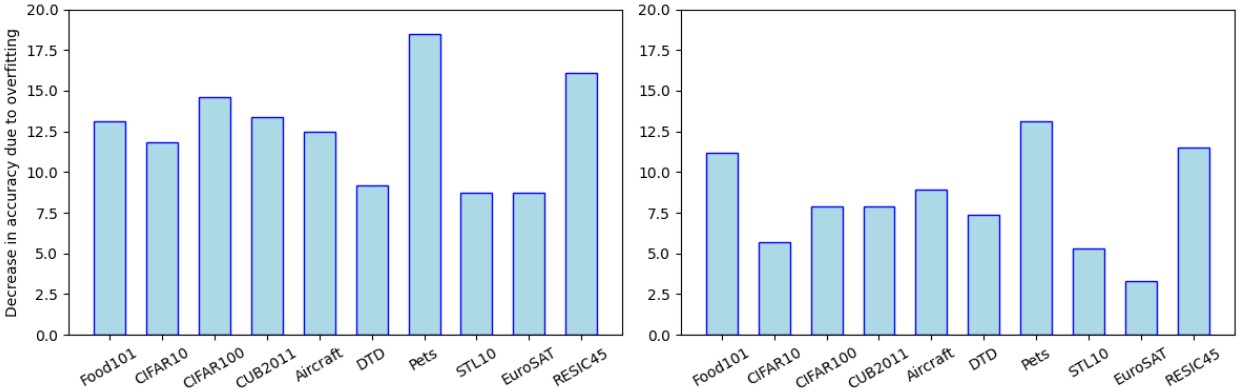

Figure 5: **Overfitting of LGSimSiam.** We visualize the advantage of using early stopping by comparing the performance of LGSimSiam at the $25^{th}$ and $6^{th}$ epoch. **Left:** Banani et al. (2023) **Right:** Ours.

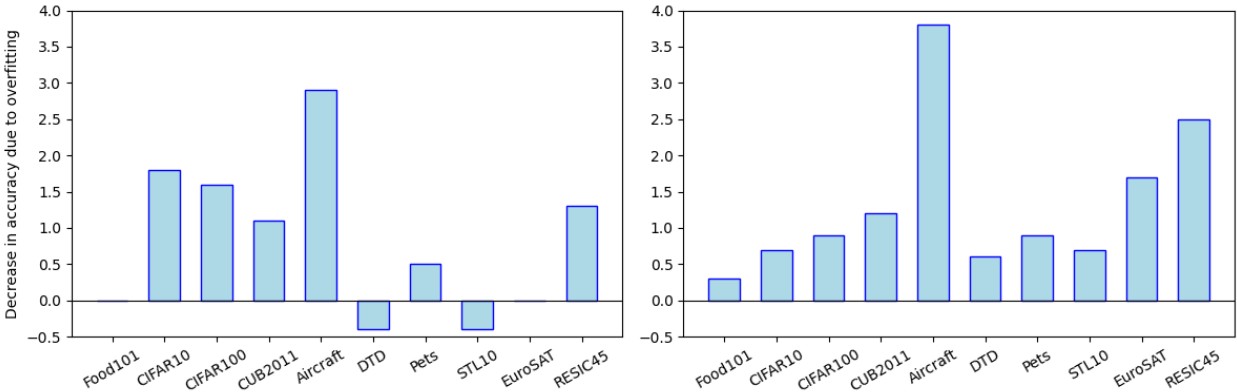

Figure 6: **Overfitting of LGSimCLR.** We visualize the advantage of using early stopping by comparing the performance of LGSimCLR at the $25^{th}$ and $9^{th}$ epoch. **Left:** Banani et al. (2023) **Right:** Ours.

## 8 Limitations

Our experiments, when compared to Banani et al. (2023), support the claim that incorporating a captioning model improves the impact of language guidance. However, the noisy and vague nature of the captions generated by the model limited the generalizability of our results. This underscores the need for a more accurate captioning model, which could potentially perform better in language-guided sampling tasks. The complete reproducibility of the original experiments could not be confirmed for three primary reasons. Firstly, the code implementation for subreddit subsampling was incomplete. Secondly, as mentioned in subsection 5.2, certain datasets could not be included in our evaluative experiments. Additionally, when we substituted the ResNet backbone with a MobileNet variant, the performance notably worsened, highlighting the backbone dependency present in the original framework.

## 9 Conclusion

Our study confirms that language guidance can improve self-supervised learning, but the effectiveness heavily depends on dataset quality, backbone selection, and training strategies. We show how early stopping is crucial when training with language guidance, as the models tend to overfit within a few epochs. Without this precaution, performance gains may be misleading or lost entirely. Additionally, dataset reliability plays a key role in the success of language guidance. Our evaluations revealed that low-quality captions can

degrade performance, while improved captions generated using BLIP-2 led to more consistent results. This emphasizes the need for careful dataset curation when using language-based methods in contrastive learning.

Furthermore, our findings highlight the importance of embedding size in self-supervised learning. The performance gap between ResNet50 and ResNet34 suggests that larger embedding space is necessary for ConvNets to capture better semantic relationships. This is critical when choosing architectures for language-guided training. Overall, our reproducibility study underscores both the potential and the limitations of language guidance in self-supervised learning. By identifying key factors like overfitting, embedding size, and dataset quality, we provide actionable insights for future research and real-world applications.

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
