# OpenReview forum: "Impact of Language Guidance: A Reproducibility Study"
_TMLR — Rejected by TMLR_

### Review · Reviewer_ioFn · 2025-04-15

**Summary Of Contributions:**

This paper examines the role of text captions in language-guided contrastive pre-training. Inspired by recent work showing that language similarity can be used to improve sampling for contrastive pre-training of visual models, the authors aim to replicate these results with some modifications to the training data and model size. The authors observe that the pre-training captions used in prior work were often of low-quality and use BLIP-2 to filter and relabel the data. Their experimental results suggest that using these generated captions may lead to improvements in a computationally-constrained setting.

**Audience:**

No

**Broader Impact Concerns:**

Reduced computational budget could provide benefit to environmental impact of language-guided sampling for contrastive pretraining of visual foundation models.

**Claims And Evidence:**

No

**Requested Changes:**

Necessary:
- Reproduce and compare with results from Banani et al.
- Demonstrate significant improvement over SOTA performance (including confidence intervals)
- Include additional details about the caption filtering strategy and demonstrate its performance on other established language-image benchmark datasets (Conceptual Captions, LAION, etc.).
- Improve motivation for saliency map, ideally with empirical results

**Strengths And Weaknesses:**

Strengths:

The authors' results are reproducible in a less computationally expensive setting than prior work

The authors note that many of the captions in the RedCaps Dataset[1] are of low quality and propose using a VLM trained on a broader language-image dataset to generate replacements for these low-quality captions. This seems well-motivated in this case and could be studied as a broader approach with a greater variety of datasets and more modern VLM architectures.

Weaknesses:

This work has very limited novelty and does not reproduce or compare to the results shown in [2], instead using the previous author's implementation with a ResNet-34 in place of a ResNet-50 backbone. On all metrics their implementation shows a decline in performance compared to the prior work [2]. While the authors motivate this as a study of "the generalization capabilities of language guided SSL to smaller, more efficient architectures such as ResNet34" they do not reproduce (or include previously published performance for) prior work using the larger backbone. As a result it's unclear whether to attribute this performance drop to the reduced model complexity or differences in their re-implementation.

The primary novelty introduced in this paper is the use of a VLM to replace captions in an existing dataset. However, even in the computationally-constrained setting considered in the experiments there is little evidence provided that this provides a significant impact for performance. Training with these VLM-generated captions does not seem to make a difference on linear probe classification (Table 2) and makes seemingly little difference on few-shot classification (Table 3). Again, because the authors do not reproduce the results in the prior work[2] it is unclear whether their ResNet-34 implementation without data filtering is a reliable baseline.

Finally, some of the metrics introduced by the authors' include little motivation or explanation. There is no definition given for the ITM score other than it "measures the alignment between an image and its caption by predicting whether they are a good match". Similarly, the saliency map metric is defined but is not well-motivated by any additional experiments to support their claim that this is a good measure of representation-learning performance.

[1] Desai, Karan et al. "RedCaps: Web-curated image-text data created by the people, for the people" NeurIPS Datasets and Benchmarks 2021
[2] Banani, Mohamed El, et al. “Learning Visual Representations via Language-Guided Sampling.” IEEE/CVF Conference on Computer Vision and Pattern Recognition, CVPR 2023

---

### Review · Reviewer_T55E · 2025-04-16

**Summary Of Contributions:**

The authors reproduce the study done by Banani et. al, 2023 for contrastive learning, where similar image representations are learned by maximizing the cosine similarity between captions.

Slightly different from the original paper, they conduct experiments on a modified dataset, which they claim has higher quality captions as they generate new captions using BLIP-2 and they use a learned filter to select the higher-scoring caption for each image.

Another difference is that they perform experiment on RESNET-34 whereas the original paper performs experiment on a RESNET-50 which has a larger dimensional feature representation.

**Audience:**

No

**Claims And Evidence:**

No

**Requested Changes:**

As mentioned above, it is unclear which experiments were conducted using ResNet-50 and which used ResNet-34.

**Strengths And Weaknesses:**

The authors have attempted to extend the study proposed by Banani et. al. through some additional experiments, but the contributions are limited and not substantial enough to warrant an acceptance.

To be precise, the only major addition is to apply the existing setup on RESNET-34, for which they claim that the results are weaker than RESNET-50. They also propose a "higher quality" dataset using an ITM based filter, but this claim is only supported by manual evaluation. Other experiments on MobileNetV3 are claimed to perform poorly and the results are not shown.

Finally, I have some comments regarding the presentation and writing style. There is a serious lack of clarity, they have not concretely mentioned what their interpretability contributions are, and apart from using Grad-CAM, it is not clear what their contribution is. I am not convinced that the omission of experiments on Caltech-101 and Oxford Flowers because of not being able to download the datasets on the author's code is justified. Moreover, information per word content of this paper is quite low.

To conclude, I believe that the paper may benefit from significantly more rigorous experiments, clearer writing, and stronger insights to build upon the original study meaningfully.

---

### Review · Reviewer_3Ryd · 2025-04-17

**Summary Of Contributions:**

This paper reproduces and extends the language‐guided sampling strategy for self‐supervised visual representation learning originally proposed by Banani et al. (2023). Its key contributions are:

1. Improving Caption Quality from noisy RedCaps captions to BLIP‑2–generated captions filtered by an Image‑Text Matching (ITM) score.
2. Backbone Optimization with ResNet‑34 and MobileNetV3, instead of ResNet-50.
3. Saliency‐map based metric (AUC‑ROC and AUC‑PR on GradCAM maps) for evaluating the semantic focus of self‐supervised ConvNets.

**Audience:**

Yes

**Broader Impact Concerns:**

Although the paper attempts to reproduce and share insights on language-guided learning, the results do not show complete evidence yet and not up-to-date. It will be useful to situate the work appropriately, systematically study the robustness and generalizability of the existing methods and then share actionable insights. Explicitly position your contributions relative to both classical SSL and modern vision–language pre-training paradigms.

**Claims And Evidence:**

No

**Requested Changes:**

Incomplete Related work: How does the work situate with respect to the recent literature in language-guided learning? Please update the related work.

ITM Filter Threshold: What score cutoff was used to select BLIP‑2 captions over originals?

Candidate Generation: How many captions per image were sampled and ranked by ITM?

Saliency‐Metric Computation: How were ground‑truth segmentation masks obtained for ImageNet‑S50? Can you explain the fig. 4 in more detail, especially why the proposed method is better?

**Strengths And Weaknesses:**

### Strengths
- Reproducing: with caption improvement pipeline and a slightly more efficient architecture choices
- Identifying Overfitting: Highlights that language-guided models tend to overfit early in training and demonstrates the importance of early stopping for accurate performance evaluation
- Insights into: the importance of embedding size, the tendency of language-guided models to overfit, and the critical role of dataset quality.


### Weakness
- Incomplete Reproducibility: No experiments with ResNet-50. Not all experiments compared and analyzed.
- Incomplete Related work: the work doesn't cite any work from 2024.

---

### Review · Reviewer_qfpU · 2025-04-17

**Summary Of Contributions:**

The authors reproduce and perform additional experiments to complement the study done Banani et al. (2023). First, they show that improving the caption quality can lead to better generalization. Moreover, they perform an ablation study on the model backbones, replacing ResNet50 from the original work with ResNet34 and MobileNetV3. Finally, the authors provide additional metrics to evaluate self-supervised learning algorithms by computing AUC-ROC and AUC-PR on saliency maps.

**Audience:**

Yes

**Broader Impact Concerns:**

If the study is more systematic, it can provide useful insights into self-supervised learning algorithms

**Claims And Evidence:**

No

**Requested Changes:**

As mentioned above, the paper would benefit from more rigorous evaluations and better writing.

**Strengths And Weaknesses:**

**Strengths**

The problem is well-motivated.
The authors manage to point out some reproducibility issue with the original paper.
Additional results using different backbones and improved caption quality are interesting.
The paper provides some useful insights into training strategies for self-supervised learning algorithms.

**Weaknesses**

In section 5.2, the author claims that “language guidance does not seem to have a significant on the quality of saliency maps”. However, from Table 1, it does seem that LGSimCLR with better captions have better AUC-ROC and AUC-PR than LGSimCLR with original captions. Moreover, I think this claim can be made more persuasive if the same observation is made across more methods and backbones.

“We investigate the generalization capabilities of language-guided SSL to smaller, more efficient architectures such as ResNet34”. It would be better if the authors can also provides results for ResNet50 to see how they compare to ResNet34.

Some of the writings lack clarity. For instance, the definition of ITM score is vaguely described.

---

> ### Author Response · Authors · 2025-05-03
>
> Regarding saliency maps: In Table 1 LGSimCLR with better captions do have slightly better AUC scores than LGSimCLR but LGSimCLR with original captions have worse AUC scores than SimCLR. Moreover the difference is little and when we visualize the saliency masks we find different models perform better across different categories (refer Fig 4). Thus we claim that language guidance does not seem to have a significance on the quality of saliency maps. We will clarify this in paper too.
> Regarding ResNet50 results: We are training on ResNet50 as well. Most of the training is complete we will share the results and add it to the paper within this week.
> Regarding clarity of presentation, the ITM score was developed by Li et. al. It is obtained using a multimodal transformer-based binary classifier, which is trained to classify whether an image-text pair matches or not. We have updated this in the paper as well.
> Thank you for your feedback, let us know if you have any further suggestions or questions.

---

### Author Response · Authors · 2025-04-03
**Inquiry Regarding Review Timeline**

Dear Action Editor and Reviewers,
When can we expect reviews on our paper? We understand that the review process can take time, and we appreciate the efforts of the reviewers and editor.
Best Regards

---

> ### Comment · Action_Editor_Xm9r · 2025-04-03
> **Thanks**
>
> It will be (at least) a few more weeks. Thanks for your patience.

---

### Decision · Action_Editor_Xm9r · 2025-06-01

**Recommendation:** Reject

**Comment:**

The authors present a reproducibility study of a previously published CVPR paper ("Learning Visual Representations via Language-Guided Sampling" by Banani-Desai-Johnson '23). Unfortunately, as all reviewers have pointed out, the study is incomplete. The experiments presented here use a smaller architecture, and while the new captioning strategies explored in this paper may be interesting, there need to be a much more thorough set of baselines and ablations needed to establish their significance. The writing in a few places is unclear and the value of the newly proposed metrics (e.g. the ITM score) is not clearly explained. Finally, CLIP/BLIP based captioning of images has been extensively explored since 2023, and for the audience to get a more complete picture, the authors might want to expand their study to include representative follow-up works to the original paper by Banani et al that have appeared since then.

**Audience:**

While the topic of the paper is well within scope, the paper needs to be substantially more fleshed out in order to appeal to a wider portion of the TMLR audience.

**Claims And Evidence:**

The paper attempts to reproduce a previous CVPR paper from 2023, but falls short in a few aspects.